

# FORMS: Forest Multiple Source height, wood volume, and biomass maps in France at 10 to 30 m resolution based on Sentinel-1, Sentinel-2, and GEDI data with a deep learning approach

Martin Schwartz[1], Philippe Ciais[1], Aurelien De Truchis[2], Jérôme Chave[3], Catherine Ottlé[1], Cedric Vega[4], Jean-Pierre Wigneron[5], Manuel Nicolas[6], Sami Jouaber[1], Siyu Liu[7], Martin Brandt[7], Ibrahim Fayad[1,2]

[1]Laboratoire des Sciences du Climat et de l'Environnement, LSCE/IPSL, CEA-CNRS-UVSQ, Université Paris Saclay, 91191
Gif-sur-Yvette, France
[2]Kayrros SAS, Paris, 75009, France
[3]Laboratoire Evolution et Diversité Biologique, CNRS, UPS, IRD, Université Paul Sabatier, Toulouse, France
[4]IGN, Laboratoire d'Inventaire Forestier, 54000, Nancy, France
[5]ISPA, UMR 1391, INRAE Nouvelle-Aquitaine, Bordeaux Villenave d'Ornon, France
[6]Office national des forêts, département Recherche-développement-innovation, Boulevard de Constance, 77300 Fontainebleau, France
[7]Department of Geosciences and Natural Resource Management, University of Copenhagen, Denmark

*Correspondence to*: Martin Schwartz (martin.schwartz@lsce.ipsl.fr)





**Abstract.** The contribution of forests to carbon storage and biodiversity conservation highlights the need for accurate forest height and biomass mapping and monitoring. In France, forests are managed mainly by private owners and divided into small stands, requiring 10 to 50 m spatial resolution data to be correctly separated. Further, 35 % of the French forest territory is covered by mountains and Mediterranean forests which are managed very extensively. In this work, we used a deep-learning model based on multi-stream remote sensing measurements (NASA's GEDI LiDAR mission and ESA's

Copernicus Sentinel 1 & 2 satellites) to create a 10 m resolution canopy height map of France for 2020 (FORMS-H). In a second step, with allometric equations fitted to the French National Forest Inventory (NFI) plot data, we created a 30 m resolution above-ground biomass density (AGBD) map (Mg ha$^{-1}$) of France (FORMS-B). Extensive validation was conducted. First, independent datasets from Airborne Laser Scanning (ALS) and NFI data from thousands of plots reveal a mean absolute error (MAE) of 2.94 m for FORMS-H, which outperforms existing canopy height models. Second, FORMS-B

was validated using two independent forest inventory datasets from the Renecofor permanent forest plot network and from the GLORIE forest inventory with MAE of 59.6 Mg ha$^{-1}$ and 19.6 Mg.ha$^{-1}$ respectively, providing greater performance than other AGBD products sampled over France. These results highlight the importance of coupling remote sensing technologies with recent advances in computer science to bring material insights to climate-efficient forest management policies. Additionally, our approach is based on open-access data having global coverage and a high spatial and temporal resolution,

making the maps reproducible and easily scalable. FORMS products can be accessed from https://doi.org/10.5281/zenodo.7840108 (Schwartz et al., 2023).



## 1 Introduction

Forests play a key role in the environment and human well-being, providing ecosystem services such as carbon sequestration, biodiversity conservation, and climate regulation (IPCC, 2019). Human activities such as deforestation,

degradation, fragmentation, and non-sustainable forest management threaten these ecosystems. To preserve these important resources, accurate and up-to-date information on forest structure, such as height, volume, and biomass, is essential for effective forest management policies. Destructive sampling has long been the only method to actually measure the biomass of a tree, which involves felling trees and weighing the biomass components (Fayolle et al., 2013; Gibbs et al., 2007; Goodman et al., 2014). More recently, terrestrial laser scanning (TLS) has emerged as a promising alternative for an accurate

estimation of tree volume without destruction (Calders et al., 2022; Demol et al., 2022; Disney et al., 2019; Liang et al., 2016) but remains limited to very few ecological research sites. These measurements are used to derive species-specific allometric equations applied to diameter and height measured routinely in the field across multiple NFI plots (Chave et al., 2005; Nogueira et al., 2008) to obtain stand-level biomass estimations. Although tree height and basal area are correlated with the wood volume used to estimate biomass, uncertainties in tree-level allometry propagate into errors when estimating

the biomass of a stand containing multiple trees (Chave et al., 2014). Forest inventories play a critical role in accurately estimating forest biomass at regional and national scales (Fang et al., 1998; Shvidenko and Nilsson, 2002) and are often used as calibration data for models used in remote sensing-based biomass estimation (Morin et al., 2019; Næsset et al., 2020). For instance, Saatchi et al. (2011) developed a global biomass map at 1km resolution based on the Geoscience Laser Altimeter System (GLAS) on the Ice, Cloud, and Land Elevation Satellite (ICESat) and trained with field measurements and airborne

LiDAR. Since 2019, the Global Ecosystem Dynamics Investigation (GEDI) mission (Dubayah et al., 2020) has been collecting high-resolution measurements of vertical forest structures through LiDAR data. Although the sampling is too sparse to derive continuous maps, this new dataset brings a tremendous amount of information on global forest structures. For instance, the GEDI Level 4 Biomass (L4B) product provides 1-km aggregated estimations of above-ground biomass density (AGBD) that come from allometric equations based on waveform metrics calibrated on the biomass measured across

forest plots (Dubayah et al., 2022; Duncanson et al., 2022). However, to properly monitor forests at a local scale, especially in Europe, where forests are divided into small stands of a few hectares, a typical 10 to 50 m spatial resolution is needed. In recent years, studies started to address this issue by spatially extrapolating GEDI height measurements with ancillary continuous satellite data such as Sentinel-1 (S1), Sentinel-2 (S2), or Landsat data, thus creating 10 to 30 m resolution height maps (Lang et al., 2022; Morin et al., 2022; Potapov et al., 2021; Schwartz et al., 2022). Additionally, the use of deep

learning, and particularly convolutional neural networks (CNNs), has brought new tools to process remote sensing data with improved accuracy and the ability to automatically learn complex multi-scale features like texture from large training datasets (Ball et al., 2017; Lang et al., 2019; LeCun et al., 2015; Liu et al., 2023; Zhu et al., 2017). Applied to GEDI data,



these models have proven increased performance compared to standard machine learning approaches (Lang et al., 2022; Schwartz et al., 2022; Fayad et al., 2023).

Here we use GEDI forest vertical structure measurements in France (more than 90 million points) with deep learning techniques to derive 10 m resolution canopy height, 30 m resolution AGBD, and wood volume density (WVD) maps of France. These products will be referred to as FORMS-H, FORMS-B and FORMS-V (FORest Multiple Source Height/Biomass/Volume) in the following. FORMS-H is computed from a U-Net deep learning model trained with Sentinel-1 (S1), Sentinel-2 (S2), and GEDI data, following the methods described in Schwartz et al. (2022). Then, we developed 75  allometric equations based on NFI data to produce FORMS-B and FORMS-V. Comprehensive validation of FORMS-H is carried out with thousands of plots from the French National Forest Inventory (NFI) data and Airborne Laser Scanning (ALS). As we used NFI for calibration, we further validated FORMS-B estimates using two independent sets of non-NFI forest plot data. Finally, we conducted a comparative analysis with other height and AGBD maps available over France to highlight the increased performances of our products. These results contribute to a better understanding of France's forest 80  structure and carbon stocks at an unprecedented spatial resolution, with potential applications in forest management, climate change adaptation, and mitigation efforts.

## 2 Data

This study relies on 15 datasets to generate and evaluate the high-resolution tree height, AGBD and WVD maps of France. Three spaceborne datasets from GEDI, S1, and S2 were employed to train the deep learning model and generate the 10 m 85  resolution FORMS-H product. To derive 30 m resolution FORMS-B/V products, we applied allometric equations based on in situ measurements from the French NFI data, along with a broadleaf/coniferous mask obtained from the Copernicus Dominant Leaf Type (DLT) map. FORMS-H was validated against several datasets, including the French NFI height data and ALS data from the French LiDAR HD campaign. Furthermore, FORMS-B/V were evaluated against two forest inventory datasets (GLORIE and Renecofor) and aggregated data at the French ecoregion scale ("SylvoEcoRegion," SER, 90  https://inventaire-forestier.ign.fr/spip.php?article773). Finally, we compared FORMS-H and FORMS-B with existing height (Liu et al., 2023; Potapov et al., 2021; Lang et al., 2022) and AGBD (Santoro and Cartus, 2023; Liu et al., 2023) products available for France. Table 1 provides comprehensive details about the datasets used in this study and how they were used to train and assess the accuracy of our FORMS products.

**Table 1: Datasets used in this paper. The column "In this study" indicates where the datasets were used in our work.**

| Product name | Type of Data | Metrics used | Date | Size and resolution | Processing | References | Data Source | In this study |
|---|---|---|---|---|---|---|---|---|



| | | | | | | | | |
|---|---|---|---|---|---|---|---|---|
| **GEDI** | Spaceborne full waveform LiDAR | L2A product v002, RH$_{95}$ height | Apr 2019-Jan 2022 | 25 m circular footprint. 234,747,773 raw footprints before and 91,537,289 (39 %) after filtration | Filters: - Quality flag=1 - Detected modes >0 - RH$_{95}$<60 m | Dubayah et al. (2021) | https://lpdaac.usgs.gov/products/gedi02_av002/#tools | Train and validation of FORMS-H |
| **Sentinel-1 (S1)** | Synthetic Aperture Radar (SAR) from the European Space Agency (ESA) Copernicus Mission | Ground Range Detected (GRD) scenes. Ascending and descending orbits: Vertical-Vertical (VV) and Vertical-Horizontal polarization (VH) | 2020-05-01 to 2020-10-01 (Leaf-on season) | 4,522 raw scenes, pixel size: 10 m | Pixel-wise temporal median for ascending and descending orbits | https://sentinels.Copernicus.eu/web/sentinel/missions | Processed and downloaded on Google earth engine (GEE) | Model training/ inference |
| **Sentinel-2 (S2)** | Multi-Spectral Imager (MSI) from ESA's Copernicus Mission | L2A product: bottom of the atmosphere reflectances: Bands (B) 2-4: Blue, Green, Red. B 5-7: Red edge, B8: Near Infrared (NIR), B8A: "narrow" NIR, B11,12: Short Wave Infrared (SWIR). | | 1,253 descending orbit and 1,203 ascending scenes Pixel size: 10 m | - Cloud content <50 % + Cloud mask - Resampling of all the bands at 10 m, - Pixel-wise temporal median | | | |
| **French National Forest Inventory (NFI)** | Field data from forest inventory measurements | Dominant height of the plot | 2020 | 5,475 circular plots (30 m) in France | | Istitut National de l'Information Géographique et Forrestiere (IGN) (https://inventaire-forestier.ign.fr/) | Requested to IGN | FORMS-H validation |
| | | WVD | | | Conversion to MgC.ha$^{-1}$ with expansion factors (ADEME and IGN, 2019) | | | Height-biomass allometry |
| **French LIDAR HD** | Airborne Laser Scanning (ALS) point-cloud data | RH$_{95}$ of height | ALS 1: 2022; ALS 2: 2021 | Point cloud data. 10pts .m$^{-2}$ at least — ALS 1: 1 large area of 2500 km² — ALS 2: 20 tiles of 1 km² | Extraction of the RH$_{95}$ height at 10 m resolution with pdal algorithm 1. Estimate height above ground (HAG) of vegetation points with *pdal hag_delaunay* filter 2. Rasterize the point cloud keeping only the max value of the HAG for each pixel at 50 cm resolution 3. Downsample at 10 m resolution by keeping the 95th percentile height | IGN (https://geoservices.ign.fr/lidarhd) | | FORMS-H validation |
| **Copernicus Dominant Leaf Type (DLT)** | Broadleaf / Coniferous map from the Copernicus Land monitoring service | Tree type for each pixel | 2018 | Pixel size: 10 m. | | © European Union, Copernicus Land Monitoring Service 2018, European Environment Agency (EEA) | https://land.Copernicus.eu/pan-european/high-resolution-layers/forests/forest-type-1 | Model training and testing (filter GEDI data) |
| | | | | 30 m | Resampling at 30 m with the dominant leaf type. | | | Production FORMS-B/V |
| **Renecofor permanent plot network** | Forest plots mainly from mature old-growth forest | DBH, Tree species, Height | 2019 | 102 plot (0.5 ha) distributed throughout France | allometric equations from Forrester et al. (2017) to infer biomass | Ulrich (1995) http://www1.onf.fr/renecofor/ | Available upon request to M.Nicolas, French forest office, (ONF) manuel.nicolas@onf.fr | FORMS-B validation |



| GLORIE forest inventory | Forest plots from a coniferous plantation in south-west of France (Les Landes) | AGBD | 2016 | 104 plots (50 m circular plots ) | | Motte et al. (2016), Zribi et al. (2019) | Available upon request to D.Guyon (INRAE Bordeaux-Aquitaine) dominique.guyon@inrae.fr | FORMS-B validation |
|---|---|---|---|---|---|---|---|---|
| **French Sylvoecoregion (SER) statistics** | Average WVD and AGBD estimations based on NFI plots statistical aggregation | WVD (m³ ha⁻¹) | 2020 | 91 SER | | | https://inventaire-forestier.ign.fr/spip.php?rubrique127 | FORMS-B/V validation |
| | | Above- and below-ground biomass density (MgC ha⁻¹) | 2014 | 40 groups of SER | Conversion to AGBD (Mg ha⁻¹) | ADEME and IGN, (2019) | https://librairie.ademe.fr/produire-autrement/808-contribution-de-l-ign-a-l-etablissement-des-bilans-carbone-des-forets-des-territoires-pcaet.html | |
| **Potapov height map** | Global canopy height map based on Landsat, GEDI, and Machine Learning. | Canopy Height | 2019 | Global map at 30 m resolution | | Potapov et al. (2021) | https://glad.umd.edu/dataset/gedi | FORMS-H comparison |
| **Lang height map** | Global canopy height map based on Sentinel-2, GEDI, and Deep Learning | Canopy Height | 2020 | Global map at 10 m resolution | | Lang et al. (2022) | https://langnico.github.io/globalcanopyheight/ | FORMS-H comparison |
| **Liu height map** | European canopy height map based on PlanetScope, ALS, and Deep Learning | Canopy Height | 2019 | European map at ~3 m resolution. | Conversion to 10 m resolution with a maximum resampling method | Liu et al. (2023) | Available upon request to to Siyu Liu (University of Copenhaghen), sliu@ign.ku.dk | FORMS-H comparison |
| **ESACCI Biomass map** | Global AGBD map | AGBD in oven dry biomass (Mg.ha⁻¹). Product version 4 | 2020 | Global map at 100 m resolution | | Santoro and Cartus, (2023) | https://dx.doi.org/10.5285/af60720c1e404a9e9d2c145d2b2ead4e | FORMS-B comparison |
| **Liu Biomass map** | European AGBD map | AGBD in oven dry biomass (Mg.ha⁻¹) | 2019 | European map at 30 m resolution | | Liu et al. (2023) | Available upon request to to Siyu Liu (University of Copenhaghen), cnliusiyu@gmail.com | FORMS-B comparison |

# 3 Methods

## 3.1 Mapping canopy height at high resolution (10 m)

To map canopy height in France at 10 m resolution, we adapted the methods developed and presented in Schwartz et al. (2022). The processing is based on a deep learning U-Net model (Ronneberger et al., 2015) adapted from (Milesi, 2022) that

learns multi-scale features in S1 and S2 images to predict canopy height. This model is trained on a pixel-wise regression



process with GEDI RH$_{95}$ height data. The RH$_{95}$ height means that 95 % of the energy returned to the sensor comes from photon reflections below this height. It is a widely used proxy for canopy height as it is less sensitive than RH$_{100}$ to atmospheric disturbances (Fayad et al., 2021; Potapov et al., 2021). Prior to the model training, France was divided into 10,000 km² areas, which we will refer to as "tiles" in the following, that we randomly split into 4408 (75 %) train, 887 (15

%) validation, and 589 (10 %) test tiles. The detailed training processes are described in Fig. 1: (1) Random selection of a train tile (2) Random selection of a 2560x2560 m subset of this tile to reduce overfitting (3) Input of the corresponding 256x256-pixel image with 14 layers from S1 and S2 (See Table 1) to the U-Net model. (4) Creation of a target height image from the GEDI RH$_{95}$ height. We used the 10 by 10 m pixel corresponding to the center of the GEDI footprint as a target (5) Calculation of the MAE loss between the model outputs and the GEDI height values (6) Loss backpropagation: Model

weights are modified according to the value of the loss gradient with respect to them. This process is a key element in the training of neural networks. We performed it here with the Adam optimizer and a learning rate of 0.01 that we decreased manually when the loss function stopped decreasing. The complete training process took approximately 24 hours and was done with the Amazon AWS cloud platform on a GPU NVIDIA Tesla T4 (16 GB).

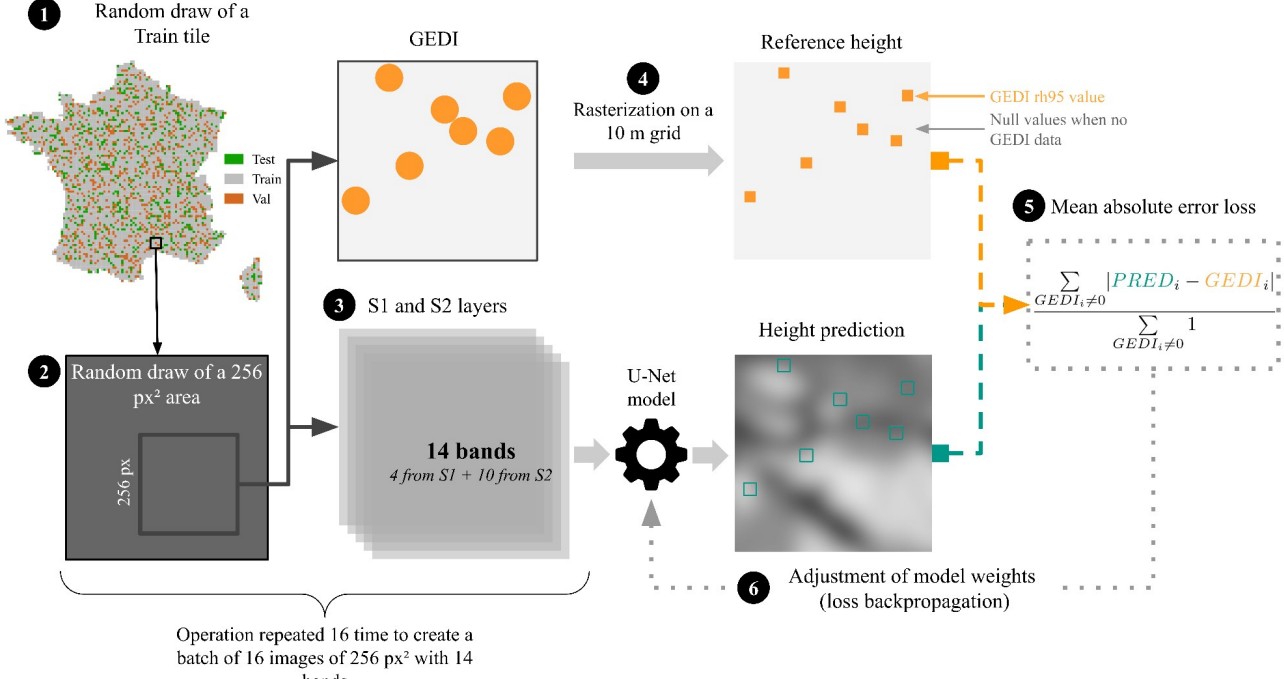


**Figure 1: U-Net training process. (1) Random draw of a train tile (2) Random draw of a 2560x2560 m subset (3) Input of the corresponding S1 and S2 layers to the model. (4) Rasterization of the corresponding GEDI data on a 10 m grid. (5) Calculation of the MAE loss (6) Loss backpropagation: Model weights are modified according to the value of the loss gradient with respect to them.**



### 3.2 From height to wood volume and biomass

To produce an AGBD map at 30 m resolution (in Mg.ha⁻¹), we derived power-law allometric equations from the French NFI plot data (See Table 1). We chose a 30 m resolution to correspond to the size of the NFI plots and to obtain a sufficient number of trees within one pixel so that AGBD has real significance. Every year, the French Geographical Institute (IGN) measures~ 6000 new plots for the French NFI (IGN, 2018). For each 30 m circular NFI plot, dendrometric measurements are made in concentric plots of 6, 9, and 15 m radius according to the DBH, and for trees having a minimum of 7.5 cm DBH. DBH and species are collected for all the trees, and height is measured for one tree per species and DBH class. The tree volumes are computed using allometric models involving DBH and height, and a WVD estimation (in $m^3.ha^{-1}$), which corresponds to the main stem's volume up to 7 cm diameter, is then derived for each NFI plot. In this study, we used ratios estimated from the CARBOFOR project (Loustau, 2010), a previous work specific to French forests, to convert the NFI wood volume to oven-dry AGBD: 0.59 Mg.m⁻³ for coniferous and 0.89 Mg.m⁻³ for broadleaved forests. These ratios combine an expansion factor that accounts for the volume of the whole tree, including branches (1.34 for coniferous and 1.61 for broadleaf) and a tree density factor (0.44 Mg.m⁻³ for coniferous and 0.55 Mg.m⁻³ for broadleaf) to obtain the oven-dry AGBD. After applying these ratios to the NFI plot WVD estimations, we compared them to the mean of the FORMS-H height in each NFI plot's 30 m circular area. Based on the dominant tree species given in the NFI data, we divided coniferous and broadleaved plots and fitted two power-law allometric equations (Fig. 2.a,b) with a Huber Regressor (Huber and Ronchetti, 1981) method that has the advantage of being less sensitive to outliers. Even though height and biomass are two different physical quantities, height-biomass power-law allometric equations have been widely used and showed satisfying results when no other variables were available to carry out biomass predictions (Enquist, 2002; Chave et al., 2005). The estimation of AGBD obtained using these power law relationships (Fig. 2.c) shows a MAE of 61.7 Mg ha⁻¹ and a R² of 0.4 when compared to the NFI AGBD. We observe a saturation for higher AGBD values > 400 Mg ha⁻¹, explained by the broad range of AGBD values observed for a given height in Fig. 2.a,b, especially for higher heights.



Earth System
Science
Data

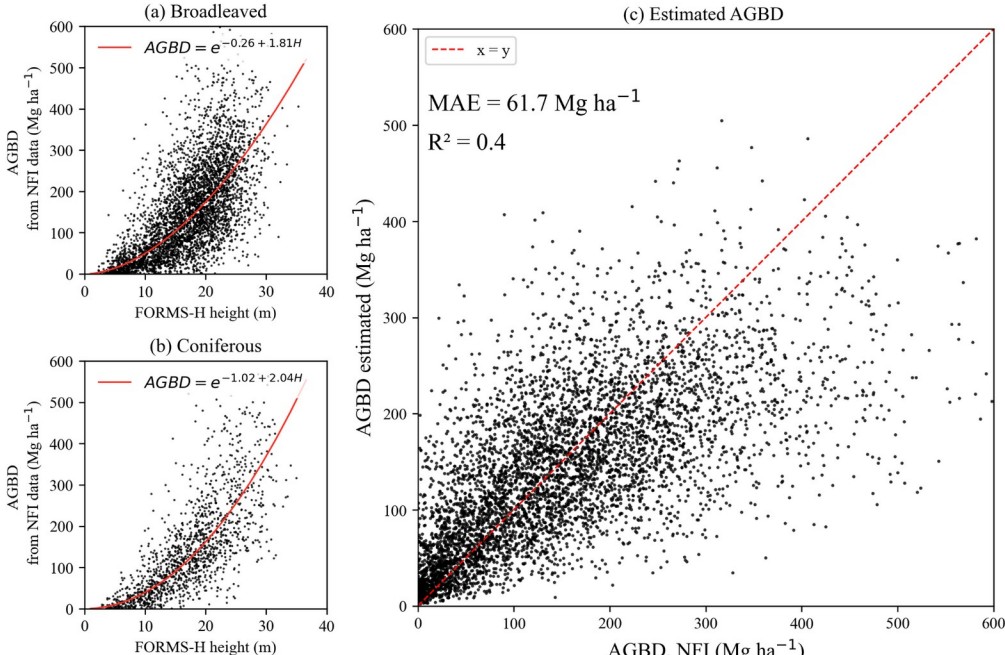

**Figure 2: (a,b) Comparison of the AGBD from NFI plot data to FORMS-H for broadleaved and coniferous forests. The red line corresponds to the power-law allometric equation that we fitted on the data and used to transform our tree height map into an AGBD map. (c) Comparison of the AGBD estimated from the allometric equations and the actual AGBD of the NFI plots. The red dashed line corresponds to the 1:1 fit.**

To obtain the AGBD map of France, we first resampled our FORMS-H height map at 30 m resolution. Then we used the Copernicus DLT map (Table 1) to estimate the fraction of broadleaf /coniferous forests within each 30 m pixel and applied these power-laws allometries (Fig. 2.a,b) accordingly to obtain an AGBD map (Mg ha$^{-1}$) of France for 2020 at 30 m resolution (FORMS-B). We also produced a WVD map (FORMS-V) at 30 m resolution (m$^3$.ha$^{-1}$) with the same method without using the volume-to-biomass ratios detailed above.

**3.3 Products validation**

To evaluate the accuracy of our 10 m resolution FORMS-H product, we compared it to four independent datasets: the 2,479,668 GEDI Test footprints, the 5,475 French NFI plots measured in 2020, and two ALS datasets (one large area of 2500 km² and one set of 20 smaller areas of 1 km² each) from an ongoing French Lidar HD campaign that aims at covering the whole national territory within the next years (Table 1). GEDI Test footprints were taken from the 589 Test tiles (Fig. 1) and filtered with the Copernicus DLT map (Table 1) to remove non-forest data. We compared here the maximum FORMS-H value within the 30 m circular plot area to the dominant height variable provided in the NFI data. This height is computed from ~ 7 representative trees and stands for the mean height of the 100 highest trees within a surface area of 1 hectare. The ALS point-cloud data come from the French LiDAR HD measurement campaign (Table 1). The first ALS site is a large



2500 km² area in the North of Paris with flat terrain (Fig. 3). The other 20 sites cover smaller areas (1 km² each) and are distributed in different sites with more complex topography. We used ALS data point cloud classification (See Table 1) to create a 50 cm resolution canopy height model that we resampled at 10 m by taking the 95th percentile of height and compared it to FORMS-H. Additionally, we conducted a comparative analysis with three height maps available globally or

in Europe to assess the novelty of FORMS-H (Lang et al., 2022; Liu et al., 2023; Potapov et al., 2021). These maps will be referred to as "Lang", "Liu", and "Potapov" respectively in the following.

We evaluated FORMS-B by comparison to two independent forest inventory datasets and to aggregated NFI statistics at a larger scale. The NFI does not provide official plot-level biomass estimates, but rather only volume and height, and our

method for establishing the height-AGBD relationships shown in Fig. 2 would lead to large errors and a degree of circularity if AGBD inferred for individual plots was used for validation. Further, it would not be an independent validation since the NFI AGBD data are used to define the relationship for transforming height maps to AGBD maps. For these reasons, we used for validation two smaller inventories ("Renecofor", "GLORIE", described hereafter) that are independent of the NFI. The Renecofor permanent plot network (Ulrich, 1995) gathers 102 forest plots distributed over France (Fig. 3) that grow under

different climatic and soil conditions. Measurements of numerous parameters, including DBH, are performed yearly by the French National Forest Office (ONF) to monitor and understand changes in forest ecosystems. These monitoring plots were installed in public forests, and their stands are managed by local foresters with the same thinning intensity as the surrounding stands. To derive a biomass estimation that we could use as reference data to evaluate our FORMS-B product, we used species-dependent DBH-based allometric equations and wood densities described in Forrester et al. (2017). We applied

generic equations based on the tree leaf type (coniferous/broadleaf) for the tree species not covered by this study. Thus, by dividing by the plot surface (0.5 ha), we obtained an AGBD estimation for each forest plot of the Renecofor network. These values were compared here to the mean FORMS-B values within a 100 m diameter circle around the location of the plot. The GLORIE forest inventory (Motte et al., 2016; Zribi et al., 2019) includes measurements of tree heights, DBH, and tree density in 104 forest stands of maritime pine located in the Landes Forest (Fig. 3). This area represents mainly private and

intensively managed forests, representative of the Les Landes area. In the GLORIE data set, AGBD estimations were derived from allometric equations applied to plot measurements of DBH (Shaiek et al. (2011) for DBH => 10 cm and Baldini et al. (1989) otherwise). These values were compared to the mean FORMS-B values within a 50 m diameter circle around the location of the plot, which corresponds to the average plot dimensions. At a regional scale, we also evaluated our FORMS-B/V products over forest ecoregions (SER, see Sect. 2). France is categorized into 91 SER based on forest types and

management practices. Every year, aggregated WVD (See Table 1) estimations are provided at the SER scale by the French forest inventory service (IGN) from the NFI plot data. To evaluate the capability of our model to carry out consistent estimation at this scale, we compared for each SER the average FORMS-V WVD estimations on forest pixels, determined with the Copernicus DLT map (Table 1), to these official data. Besides, in a report evaluating the French carbon footprint by ADEME and IGN in 2019, France's above and below-ground carbon densities were assessed for groups of these SERs in



2014. These estimations, given in above- and below-ground carbon per hectare, were converted to AGBD with ratios described in Loustau et al. (2010): the values were divided by 0.475 MgC.Mg$^{-1}$ to estimate the oven-dry biomass and by a 1.3 root expansion factor to account for AGB only. As for FORMS-V, we further compared these estimates to FORMS-B to assess its performance at a regional scale. Additionally, we compared FORMS-B to two other biomass products available globally or for Europe (Liu et al., 2023; Santoro and Cartus, 2023), which were converted when necessary to WVD maps

with the factors detailed in Sect. 3.2. These maps will be referred to as "Liu" and "ESACCI" respectively in the following.

In this study, we used several error metrics, including the mean absolute error (MAE), the mean error (ME), the mean absolute percentage error (MAPE), and the coefficient of determination (R²). The MAE gives information about the overall error, the ME highlights the model's bias, and the MAPE computes the relative error percentage to compare the model's

performances on different validation datasets. We applied MAPE only to heights > 5 m and to AGBD > 10 Mg ha$^{-1}$ to avoid infinite values and to evaluate our model's performances solely on forests. The R² score indicates the performance of a regression task. A score of 1 indicates that the predicted values perfectly fit the reference data. A score below zero indicates that the model performs worse than a model predicting the average value. The detailed formula of these metrics can be found in Appendix A.



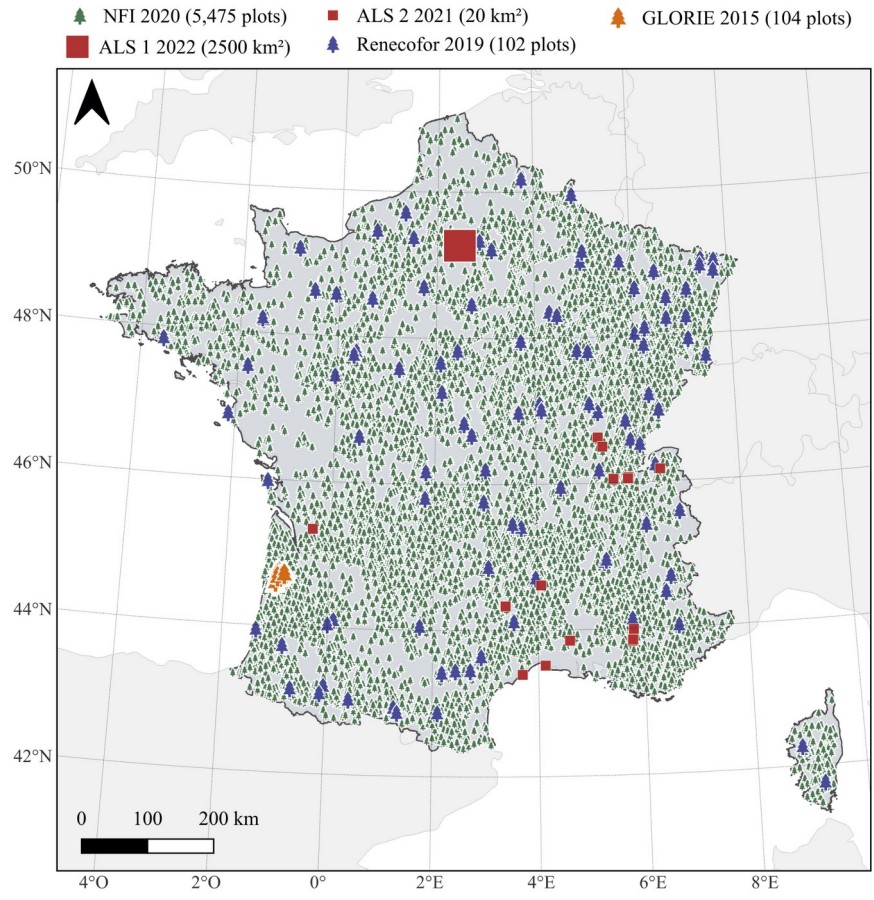


**Figure 3: Presentation of the study site with the French NFI plots (green), the Renecofor forest plots (blue), the GLORIE forest plots (orange) and ALS 1, ALS 2 data (red) used for validation.**

## 4 Results

### 4.1 FORMS-H: France canopy height map at 10 m resolution (2020)

Our 2020 FORMS-H product for France at 10 m resolution is presented in Fig. 4.a. The overall picture highlights the fragmentation and the variety of forest types in France with forest heights mainly ranging from 0 to 30 meters. The details of height prediction presented in Fig. 4.b show the ability of our map to retrieve precisely forest landscape units visible on Google Maps on a broad range of heights. Forest parcels are distinctly visible with precise borders.



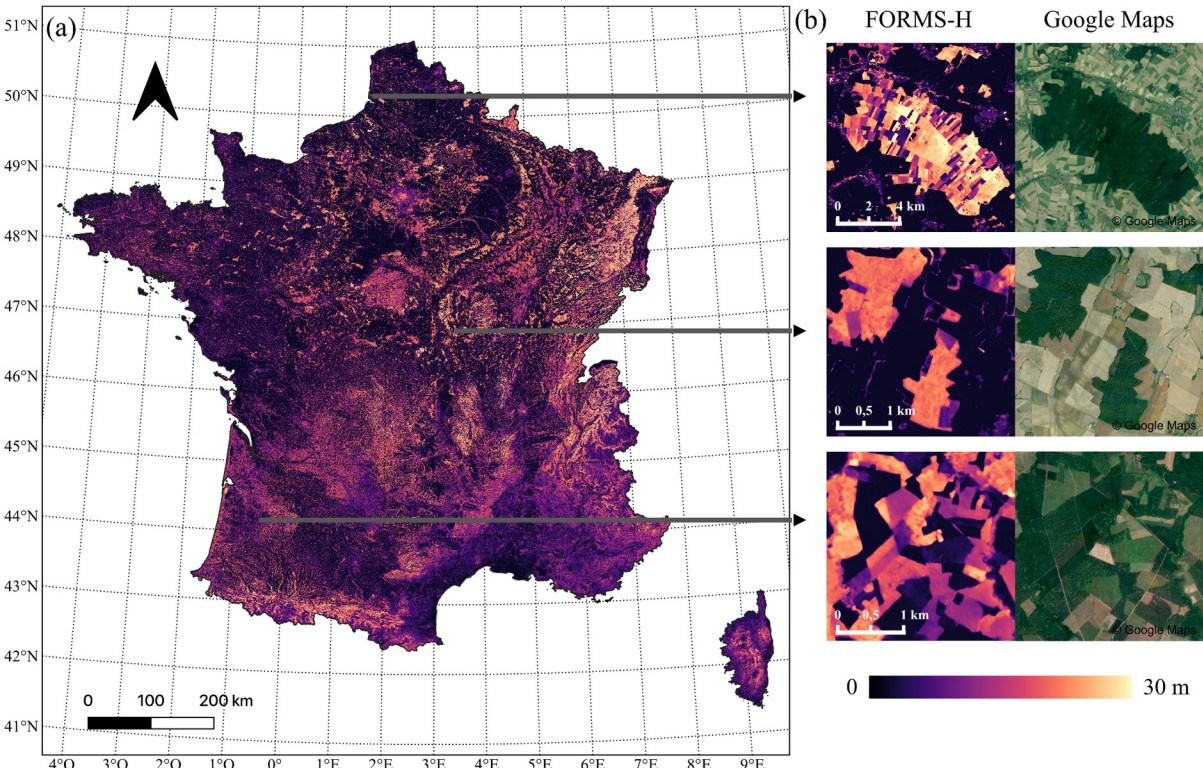

**Figure 4: (a) FORMS-H: Tree height map of France at 10 m resolution for the year 2020. (b) Examples at three different locations of height prediction (left) with the corresponding Google map images (right). Brighter colors indicate higher heights.**

We evaluated FORMS-H using four different datasets, including the GEDI Test $RH_{95}$ (Fig. 5.1) height data, French NFI dominant height data collected in 2020 (Fig. 5.2 ), and two ALS datasets from the French LiDAR HD campaign (Fig. 5.3,4). The comparison with the Test GEDI footprints (See 3.1) yields an MAE of 4.48 m and R² of 0.33 (Fig. 5.1.a). This initial validation step demonstrates the ability of our model to correctly map the GEDI $RH_{95}$ variable for a broad range of heights. Figure 5.1.b boxplots highlight the high precision of our model with a low bias, especially in the 5-30 m height range (ME = 4.8 m (5-10 m), ME = - 0.2 m (15-20 m), ME = -3.8 m ( 25-30 m)). Higher heights are more challenging to predict, and the model tends to underestimate them (ME = -6.7 m for the 30-35 height range). Conversely, FORMS-H indicates higher heights than the labeled GEDI footprints for many areas categorized as low heights (Figure 5.1.a). This discrepancy can likely be explained by the quality of GEDI data, where the labels could be wrong due to atmospheric conditions or geolocation errors. Our comparison with the completely independent French NFI data (Fig. 5.2) excludes these types of outliers as it focuses only on forests measured in 2020. It yields a smaller MAE of 2.94 m (Fig. 5.2.a) with a distribution of predicted data very close to the NFI distribution of heights (Fig. 5.2.b). Similarly to the comparison vs. the GEDI Test data set, the boxplots show that higher heights above 25 m tend to be underestimated with a ME of -2.8 m for the 25-30 m range and - 4.5 m for the 30-35 m range of heights. The performances of the model are slightly better in coniferous forests (MAE =





2.93 m, R² = 0.74) compared to broadleaved forests (MAE = 2.94 m, R² = 0.65, See Appendix B). The validation with ALS 1 (Fig. 5.3) and ALS 2 (Fig. 5.4) data from the French LiDAR HD campaign (Table 1) confirms the conclusions obtained from the previous datasets. ALS 1 comparison (Fig. 5.3.a) yields a result similar to the French NFI with an MAE of 3.54 m, R² of 0.61, and a comparable underestimation of higher heights (ME = -3.36 m for the 25-30 height range). Lower heights

240    tend to be overestimated (ME = 6.47 m for the 0-5 m height range) with some low-height pixels that our model predicted much higher. This discrepancy can be attributed to forest borders, where FORMS-H has smooth height transitions while the ALS data capture sharp edges more effectively. The ALS 2 data comes from a more diverse range of terrains and locations but still yields a good correlation with an MAE of 4.51 m and R² of 0.53. However, we observe an increased tendency to underestimate high height values (ME = -6.13 m for the 25-30 height range) and overestimate low values (ME = 4.82 m for

245    the 0-5 m height range). Notably, the peak of height distribution at ~17 m in the histogram of Fig. 5.4.b is 1.8 times higher than in the ALS 2 distribution, indicating that the model tends to predict the mean value rather than spanning the whole distribution of heights.



**Figure 5: Comparison of FORMS-H with four reference datasets. The figure displays scatterplots (a) and histograms with box plots (b) for each of the four datasets: (1) GEDI TEST RH$_{95}$ height data, (2) French NFI height from 2020 measurements, and (3-4) two ALS RH$_{95}$ heights from French LiDAR HD measurements. Only forest pixels from the DLT map (See Table 1) are shown. In each comparison, the scatterplot (a) shows a density plot of the predicted height plotted against the reference height, with brighter colors indicating a higher density of points. The dashed line represents the 1:1 axis. The histograms with box plots (b) display the differences between the predicted and reference height for each height range of 5 meters. The median value is represented by a red line, while the upper and lower quartiles are represented by the upper and lower edges, respectively. The whiskers symbolize the 5$^{th}$ and 95$^{th}$ percentiles.**

We conducted a comparative analysis between FORMS-H and three other existing tree height maps sampled over the entire metropolitan French territory (See Sect. 3.3). A visual comparison of the three maps (Fig. 6.a,b,c,d) highlights the improved ability of FORMS-H to reproduce spatial patterns visible on Google Maps and ALS data compared to the Lang and Potapov maps. FORMS-H performed well in flat terrain (Fig. 6.a,b) as well as in areas with higher slopes (Fig. 6.c,d, slope ~30°).

30



Lang's map, released at a spatial resolution of 10 meters, appears to be coarser than FORMS-H, although it still captures some height patterns observed in ALS data. In contrast, the Potapov map failed to capture most of these patterns. The Liu's map, based on PlanetScope data (3 m resolution) for predicting heights after training with ALS data in selected European areas (all outside France), captures the spatial heterogeneity well and follows the pattern observed in ALS data. Furthermore,

individual tree crowns and very fine-scale landscape units are visible in this very high-resolution height dataset. To verify our visual comparison, we further quantitatively compared FORMS-H to ALS data resampled at 10 m resolution (Fig. 6.e,f) and found that it outperformed the other models significantly for Lang and Potapov, with an MAE of 3.54 m (ALS 1) and 4.5 m (AL2) compared to an MAE of 4.97 m (ALS 1) and 5.57 m (ALS 2) for Lang, and 5.72 m (ALS 1) and 6.8 m (ALS 2) for Potapov. $R^2$ coefficients confirm the superiority of FORMS-H (ALS 1: 0.61, ALS 2: 0.53 ) compared to Lang (ALS 1:

0.27, ALS 2: 0.24) and Potapov (ALS 1: 0.07, ALS 2: -0.12 ). Interestingly, FORMS-H also performs better than the Liu map resampled at 10 m resolution for ALS 1 (MAE = 4.76 m, $R^2$ = 0.41) and similarly for ALS 2 (MAE = 4.53 m, $R^2$ = 0.50), even though this model was trained on higher resolution images, with ALS as reference data, which is more precise but with an uneven spatial distribution and no data in France compared to our GEDI reference data. All maps tend to underestimate the higher heights measured by ALS.

Earth System
Open Access Science
Data Discussions




**Figure 6: Visual (a,b,c,d) and quantitative (e,f) comparison of ALS RH$_{95}$ height data resampled at 10 m resolution to FORMS-H (10 m resolution) and to three other products at 3 m (Liu et al., 2023), 10 m (Lang et al., 2022) and 30 m resolution (Potapov et al.,**



2021). (a) comes from ALS 1 data while (b,c,d) are in ALS 2 data. (a,b) are located on flat terrains. (c,d) are located on steep
terrain with 20° to 40 ° slopes. (e, f) show the comparison with all ALS 1 and ALS 2 data in forest pixels filtered with the DLT map
(Table 1). The black dashed line is the 1:1 axis. We applied a uniform noise value in the [-0.5, 0.5] range to the three other height
products to allow a better scatterplot density visualization due to the data type provided as integers without changing the
performance metrics.

## 4.2 FORMS-B/V: France AGBD and WVD maps at 30 m resolution (2020)

Based on the power-law allometric equations fitted between NFI AGBD and our height estimates for each NFI plot, as
described in Sect. 3.2, we derived FORMS-B (Fig. 7.a), a 30 m resolution AGBD map of France in Mg ha$^{-1}$, and FORMS-V,
a 30 m resolution WVD map of France in m$^3$ ha$^{-1}$. The different colors on the map represent varying levels of AGBD, with
brighter colors indicating higher values.

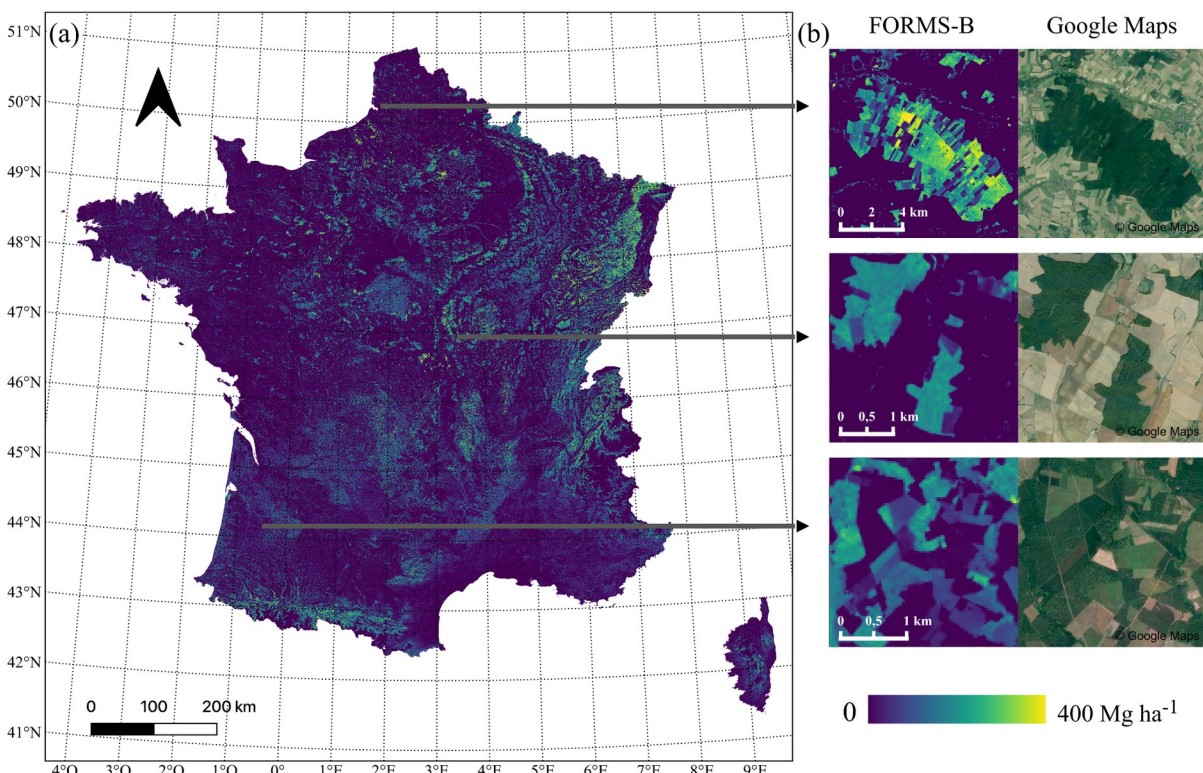

**Figure 7: (a) FORMS-B: AGBD map of France at 30 m resolution for 2020. (b) Examples at three different locations of biomass
prediction (left) with the corresponding Google map images (right). Brighter colors indicate higher AGBD.**

To assess the performance of the 30 m resolution FORMS-B product, we compared it to two existing wall-to-wall biomass
maps from global and European studies, sampled over France (See Sect. 3.3). Even though the three maps were obtained
with different data sources and methods, they all mostly agree on biomass order of magnitudes and high biomass locations

35                                                                 18





(Fig. 8.a,b,c,d). Moreover, FORMS-B, along with the Liu map, has a higher resolution, enabling more precise biomass estimations at the forest parcel level, whereas the global product ESACCI has a coarser resolution (100 m) that precludes detailed analyses. To gauge the accuracy of each map, we further quantitatively compared them to the two forest Renecofor and GLORIE inventory datasets (Fig. 8.e,f, Sect. 3.3). The Renecofor forest plots mainly consist of mature forests,

considered more challenging for biomass estimations, due to the presence of understory trees not evidently sampled from satellites. FORMS-B performs better on this "Challenging test" dataset than the two other biomass maps, with an MAE of 59.7 Mg ha$^{-1}$, (vs 63.7 Mg ha$^{-1}$ for Liu and 90.7 Mg ha$^{-1}$ for ESACCI) for biomass values reaching up to 430 Mg ha$^{-1}$. Similarly as for FORMS-H, the FORMS-B product ($R^2$ = 0.18) has a close performance to the Liu map ($R^2$ = 0.17). FORMS-B is based on two height-biomass allometric equations for coniferous and broadleaved trees and cannot capture

specificity due to different tree species, which partly explains the under- or over-estimations of biomass for individual forest plots. The comparison analysis with Renecofor plots for the two other biomass maps leads to the same conclusion but with a higher MAE. The GLORIE forest plots are all located in the Landes Forest, an intensive maritime pine plantation in the South West of France. On this dataset, with AGBD ranging from 0 to 160 Mg ha$^{-1}$, FORMS-B has an MAE of 19.7 Mg ha$^{-1}$, better than the two other products with an MAE of 26.5 Mg ha$^{-1}$ and 28.3 Mg ha$^{-1}$ for Liu and ESACCI respectively. We

observe a tendency to overestimate lower AGBD values which can be explained here by the time difference between the date of inventory and the date of the maps presented here, given the high growth rates of young maritime pines in the study area (Lemoine, 1991). Overall, Fig. 8 highlights the capability of FORMS-B to estimate biomass density, compared to other existing biomass maps, across various types of forests in France. Still, errors are larger (MAPE = 24.1 % for Renecofor, 39.9 % for GLORIE) than the ones obtained for the height map validation (MAPE = 18.4 % for the NFI plots, 16.9 % for ALS 1

and 22.5 % for ALS 2), which showcases the difficulty to derive an AGBD map only from a height-based product. Our height-biomass allometric equations for both coniferous and broadleaved forests could be refined with other parameters such as forest cover and detailed for specific species, which requires a high-resolution dominant species map to increase FORMS-B accuracy.





40




**Figure 8: Visual (a,b,c,d) and quantitative (e,f) comparison of FORMS-B to two other products (Liu et al., 2023; Santoro and Cartus, 2023). (a,b) are located on flat terrains. (c,d) are located on steep terrain with slopes from 20° to 40 °. (e) shows the comparison with the biomass from Renecofor forest plots (2019), including mainly old-growth mature forests uniformly distributed over France. (f) shows the comparison with maritime pine forest plots (2016) from the GLORIE project in an intensively managed forest (Les Landes, southwest of France). Points circled in red represent outliers related to clear-cuts between**
**the date of the inventory and 2020 and were removed from the calculation of error metrics. The red dashed line represents the 1:1 axis.**

The agency responsible for the French NFI, IGN, produces yearly statistics of wood volume and sometimes biomass at different scales from the statistical aggregation of French NFI plots. Here we compared the 2020 WVD statistics and the 2014 AGBD statistics (most recent AGBD available statistics) to our FORMS products and to Liu and ESACCI maps at the

Sylvo-Eco-Regions (SER) level (See Sec. 3.3). The AGBD maps were converted to WVD with the ratios defined in Sect. 3.3. Figure 9 shows these comparisons for WVD (a) and AGBD (b), where each point represents a SER or a SER group. Both for WVD and AGBD, our FORMS products are closer to the NFI aggregated values. For wood volume, FORMS-V has a MAE of 30.0 $m^3$ $ha^{-1}$, which is significantly smaller than other maps (Liu: 40.0 $m^3$ $ha^{-1}$, ESACCI: 55.8 $m^3$ $ha^{-1}$). For AGBD, FORMS-B (MAE = 19.4 Mg $ha^{-1}$) and Liu's map (MAE = 22.4 Mg $ha^{-1}$) have similar performances that outperform the

global ESACCI (MAE = 38.7 Mg $ha^{-1}$) map. All the products underestimate the average WVD (ME for FORMS-V: -27 $m^3$ $ha^{-1}$, Liu: -25 $m^3$ $ha^{-1}$ , ESACCI: -48 $m^3$ $ha^{-1}$) and AGBD (ME for FORMS-B: -15 Mg $ha^{-1}$, Liu: -12 Mg $ha^{-1}$ , ESACCI: -35 Mg $ha^{-1}$) of forests at SER scale. Fig. 8.e showed that all the products underestimated large AGBD values in Renecofor, which could explain this underestimation at an aggregated regional scale. Additionally, we computed the WVD and AGBD averages on all the forest pixels from the DLT Copernicus map, which includes areas not considered as forests by IGN and

that could have lower AGBD and WVD values.

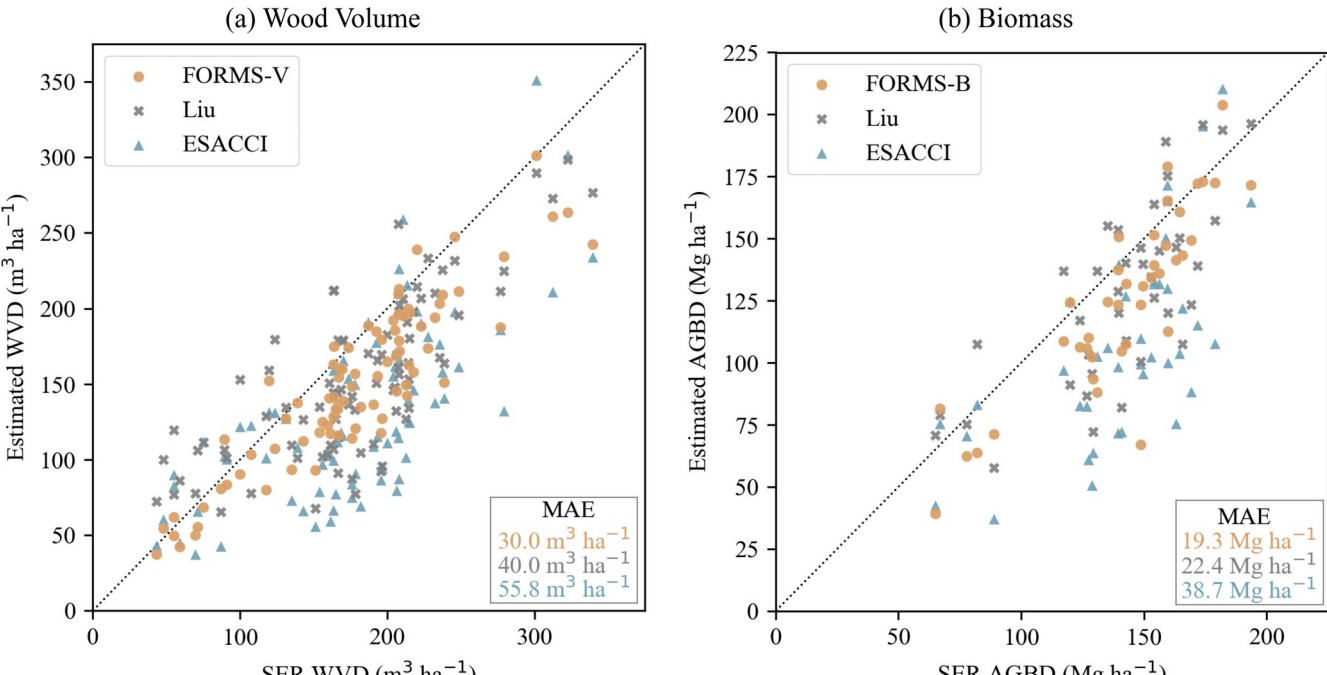

(a) Wood Volume            (b) Biomass



**Figure 9: (a) Comparison of FORMS-V, Liu, and ESACCI average WVD with the WVD disclosed in 2020 NFI statistics for the 91 French SER. The AGBD products were converted into wood volume with ratios described in Sect. 3.3. The average WVDs were estimated for forest pixels only according to the Copernicus DLT map (Table 1) (b) Comparison of FORMS-B, Liu, and ESACCI average AGBD with 2014 AGBD estimations (ADEME and IGN, 2019) for 40 groups of SER. The reference SER data were converted from above and below-ground carbon to oven-dry AGB as described in Sect. 3.3. The average AGBDs were estimated for forest pixels only according to the Copernicus DLT map (Table 1). The black dotted lines represent the 1:1 axis.**

## 5 Data availability

FORMS products presented in this paper can be visualized online at https://martinschwartz0.users.earthengine.app/view/forms-height-biomass-volume-viewer. They can be directly used as Earth Engine images datasets named "projects/ee-martinschwartz0/assets/FORMS-H", "projects/ee-martinschwartz0/assets/FORMS-B", and "projects/ee-martinschwartz0/assets/FORMS-V") or downloaded from the Zenodo online platform under https://doi.org/10.5281/zenodo.7840108 (Schwartz M., Ciais P., De Truchis A., Chave J., Ottlé C., Vega C., Wigneron JP., Nicolas M., Jouaber S., Liu S., Brandt M., & Fayad I. (2023). FORMS: Forest Multiple Source height, wood volume, and biomass maps in France at 10 to 30 m resolution based on Sentinel-1, Sentinel-2, and GEDI data with a deep learning approach. [Data set]. Zenodo. https://doi.org/10.5281/zenodo.7840108).

The availability of all the datasets used in this study is presented in Table 1. GEDI, S1, S2, HD LiDAR, Copernicus DLT map, SER statistics, Potapov, Lang, and ESACCI maps are open access and freely available by following the links mentioned in Table 1. Other datasets were either protected by privacy rules or made available upon request to their owners. These dataset providers were added as co-authors of this paper as their work largely contributed to the creation and validation of our products.

## 6 Conclusion

In this study, we produced three maps that bring material information on French forests for 2020. First, a canopy height map of France at 10 m resolution (FORMS-H) based on a novel deep learning approach that combines GEDI, S1, and S2 data and extends the previous work from Schwartz et al. 2022 on the forest of Les Landes. This map outperforms existing canopy height estimations over France compared to reference ALS data. Then, we produced a WVD (FORMS-V), and an AGBD (FORMS-B) maps at 30 m resolution, resulting from applying power law allometric equations to FORMS-H. Similarly, compared to other available products, these maps show a better agreement with field data and pave the way towards a fine-scale biomass monitoring of French forests. Such maps could be produced yearly and integrated into the NFI data, thus following the guidance of the Global Forest Observation Initiative (GFOI) to integrate earth observation data into national forest monitoring systems. Furthermore, our approach could be used to derive annual maps to monitor changes in forest height and biomass and serve as a reliable baseline for forest monitoring.



# 7 Appendices

## Appendix A: Error metrics


$$MAPE = 100 \cdot \frac{1}{n} \sum_{i=1}^{n} \left| \frac{e_i - t_i}{t_i} \right|$$

$$MAE = \frac{1}{n} \sum_{i=1}^{n} \left| e_i - t_i \right|$$

$$ME = \frac{1}{n} \sum_{i=1}^{n} e_i - t_i$$

$$R^2 = 1 - \frac{\sum_{i=1}^{n} (t \textrm{¿¿} i - e_i)^2}{\sum_{i=1}^{n} (t \textrm{¿¿} i - \bar{t})^2 \textrm{¿}} \textrm{¿}$$

Where $e_i$ is the $i^{th}$ estimated value, $t_i$ the $i^{th}$ true value, $\bar{t}$ the mean of $t_i$ values, and n the sample size.

## Appendix B: NFI broadleaved/coniferous forests height validation

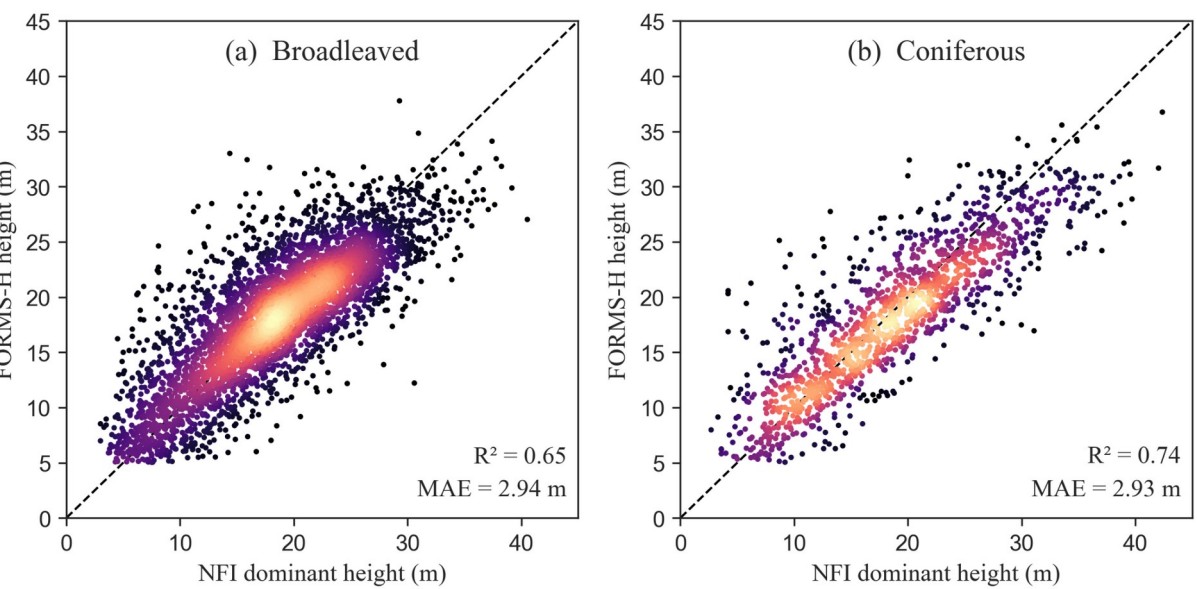

**Figure B1: Comparison of FORMS-H to French NFI heights for broadleaved (a) and coniferous (b) forest plots measured in 2020. Brighter colors indicate a higher density of points. The dashed line represents the x = y axis.**



## 8 Author contribution

MS and PC designed the study. MS developed the model and processed the data. AT and IF contributed to the model developments. CV, JC, MN, SJ, SL  provided the datasets used in this study. All authors contributed to write the manuscript.

## 9 Competing interests

The authors declare they have no competing interests.

## 10 Acknowledgments

We gratefully acknowledge Domnique Guyon and Sylvia Dayau (INRAE Bordeaux-Aquitaine) for providing the GLORIE forest inventory data, which was used to evaluate FORMS-B AGBD estimations.

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
