# Peer review of "FORMS: Forest Multiple Source height, wood volume, and biomass maps in France at 10 to 30 m resolution based on Sentinel-1, Sentinel-2, and GEDI data with a deep learning approach"

_Earth System Science Data, 2023_

## Author Comment (AC1)

**Answers to RC1**

*RC1 original text is italicized and grey*. Our answers are in plain text and black

*Summary*

*Accurate forest height and biomass mapping and monitoring is important for forest management and biodiversity conservation. Here Schwartz et al., generated a 10 m resolution canopy height map in 2020, by integrating multis-source remote sensing dataset and a deep-learning model; subsequently, with allometric equations fitted to nation forest inventory (NFI), they generated a 30 m resolution above-ground biomass density (AGBD) map. The fine resolution from 10 m to 30 m is essential for analyzing forests in France, which are typically divided into small stands. Through extensive validation against multi-source independent and observational dataset, they showed greater performance for their generated dataset compared to existing canopy height and AGBD products. The manuscript is generally well organized and well written, and the research is important. Here, I listed a few concerns regarding the manuscript.*

Thank you for this nice summary of our work. We will try our best to address your concerns in the revised manuscript.

*Specific comments*

*1)      Line 104-105, does the randomness of the split affect the model performance? Generally, in computer science and Earth science, such random split will be repeated for a few times. The mean and standard deviation of the performance metrics derived from a few experiments will be used to show the model performance and related uncertainty.*

Thank you for your comment. We rely in this study on a 1-fold cross-validation, meaning that we only divided our data into Train, Validation, and Test dataset only once. Ideally, it would have been better to do an N-fold cross-validation with N greater than one to evaluate how the randomness of the split affects the model performances and obtain the mean and standard deviation of the performance metrics. However, for extensive datasets like the one used in this study (~100,000,000 GEDI footprints), we can argue that two random splits must have similar properties. Additionally, the computational costs of model training (~24 h) make it complex to repeat this process several times. Finally, we used several external validation sources (ALS and NFI height data), which are in good agreement with the GEDI Test validation and give a good idea of the performance metrics of the model on independent validation datasets.

*2)      Line 108-109, "We used the 10 by 10 m pixel corresponding to the center of the GEDI footprint as a target". It seems that the spatial resolution of the input data is 10 m, but the output GEDI data has a resolution of 25m, the sub-pixel (i.e., across 10 m grid cells) heterogeneity within each GEDI footprint should not be contained in the output data. Also the NFI data has a resolution of 30m, then how to validate that the generated canopy height data at 10 m resolution captured the heterogeneity at that scale? Why not unify the input data to the same resolution (e.g., 30m) of GEDI or NFI or generated AGBD?*

We fully agree that this scale issue could bring uncertainties to our model. All studies that use data fusion face the same type of issues as the shape and scales of different data never match perfectly. In

our study, GEDI is used only to train the model. When we produce FORMS-H, we use S1 and S2 images. Thus, the uncertainty brought by the scaling issue only adds label noise to the data, especially for heterogeneous canopies. In more homogeneous forest cover, this effect should be reduced.

We checked if the model tends to "smooth" its predictions to handle this noise, thus creating "false" 10 m resolution maps. To do so, we designed a modified U-Net model (the "modified model" in the following) where a 2x2 convolution with stride 2 has replaced the last layer. Because of this final layer, the output of the modified model is now at a 20 m resolution. We trained this model with GEDI data rasterized on a 20 m grid, hence reducing the scaling issue. The following figure describes this model and shows the 20 m resolution GEDI target.

[Figure]

*Figure 1: Details of the original and modified models' two last layers. In the modified model, the last 1x1 convolution is replaced by a 2x2 convolution with a stride value of 2, hence creating a 20 m resolution output that can be compared to GEDI footprints rasterized at 20 m to train the model. The reference height data used for the 20 m resolution model are much closer to the footprint size of GEDI than the one used for the original model, hence addressing the scaling issue.*

To proceed to a fair comparison with the output of our model that has a native pixel size of 10 m, we applied a bilinear upsampling to the output of the modified model, thus creating a 'super-resolution' map with 10 m pixels.

[Figure]

*Figure 2: Workflow proposed to compare the model trained with GEDI footprints rasterized at 20 m, and the model that used GEDI footprints rasterized at 10 m. The output map at 20 m is resampled at 10 m in order to create a 10 m map for a fair comparison to our original product.*

To avoid long training times, we restricted our training to a maritime pine plantation in southwest France called the Landes de Gascogne (~ 130,000 GEDI footprints). We trained a first model as we did for FORMS-H ("original U-Net") and a second one as described above ("modified U-Net"). We can visually see that the original 10 m model is more able to retrieve heterogeneous canopy cover, despite the fact that GEDI footprints have a resolution closer to 20 m. As an illustration, in the following examples, the red squares indicate holes in the canopy that the modified model at 20 m could not capture.

[Figure]

*Figure 3: Visual comparison between Sentinel-2 images, our model trained at 10 m resolution and the modified model trained at 20 m resolution here upsampled at 10 m for a fair comparison. Complex forest structures in the canopy indicated by red squares are not well retrieved in the modified model outputs, while our model can retrieve them.*

Based on these elements, we are confident that the label noise created by the scaling difference between GEDI and Sentinel is well handled by the deep learning model. Heterogeneous canopy height is well retrieved by the model at 10 meters, and the use of GEDI at a 10 m scale yields better results than at a 20 m scale, even if 20 m is closer to the original 25 m GEDI footprint shape. Therefore we think that, even though there is a scale and shape difference between GEDI footprints and S1-S2 pixels, it is possible to train the model with GEDI sampled at a 10 m scale. To our understanding, this is mostly because of the U-Net model that is able to cope with label noise.

In the methods section of the revised manuscript, we will mention this scaling difference between Sentinel 10 m resolution images and GEDI 25 m footprints. We will explain our attempt to use GEDI at a coarser scale, more consistent with the physical signal measured, that led to a decline in model performance.

We acknowledge that the comparison with 30 m circular plots from the French NFI data cannot validate if our map captures real forest heterogeneity at 10 m. However, in Fig.5, we show in the comparison of FORMS-H with two high-resolution ALS images that FORMS-H captures heterogeneity at the 10 m scale.

*3) Line 111-112, the loss function should be the loss on the validation dataset, right? Please clarify it. To make sure the results reproducible, it could be better to list the learning rate used. In addition, are there any strategies used to avoid overfitting of the trained models?*

Thank you for this comment. During the validation phase, after each training epoch, the loss function is applied to the outputs of the model applied to images from the validation dataset. It allows us to follow the evolution of the model's performances on an independent set of data. When this validation loss stopped decreasing, we decreased the learning rate by a factor of ten (after ~20 hours of training). We will modify the manuscript to add this information.

As stated in line 107, when a tile is selected to be part of a training batch, we only take a random 256 x 256 pixel subset of this tile, making $(1000-256+1)^2 = 555,025$ different possibilities for each tile. This process reduces model overfitting as the input images of the model are always different, even if they come from the same tile.

*4) Line 133-134, "we compared them to the mean of the FORMS-H height in each NFI plot's 30 m circular area". For the finally generated dataset, how did you upscale from 10 m to 30 m resolution? First calculate the mean FORMS-H height within each 30m grid cell, then calculate its corresponding AGBD or wood volume? Please clarify it in the main text. Then again, why not generate the canopy height data at 30 m resolution during the first step?*

The method to generate the final dataset at 30 m resolution is described in lines 147-151: "To obtain the AGBD map of France, we first resampled our FORMS-H height map at 30 m resolution. Then we used the Copernicus DLT map (Table 1) to estimate the fraction of broadleaf /coniferous forests within each 30 m pixel and applied these power-laws allometries (Fig. 2.a,b) accordingly to obtain an AGBD map (Mg ha-1) of France for 2020 at 30 m resolution (FORMS-B)." To upscale to 30 m resolution, we took the mean height within each 30m grid cell. This information will be added to the revised version of the manuscript.

The FORMS-H dataset has a resolution of 10 m and was not generated at 30 m resolution during the first step. We chose to keep this resolution as the level of detail for height is higher at 10 m, enabling us to accurately describe gaps in the canopy and forest edges that would not be visible at a lower resolution (See answer to comment #2). However, for AGBD and WVD, a 30 m resolution seems more appropriate and is needed due to the way we built the allometric equations with NFI plots.

*5)      Line 150-151, so you fitted FORMS-H height against NFI WVD for the final WVD data generation, right? Please clarify it. Since NFI WVD and NFI AGBD have a linear relationship (i.e., linked through the volume-to-biomass ratio), the fitted non-linear relationship between AGBD-height and WVD-height should be the same except for a scaling factor, correct? It could be better to put the fitted results of WVD-height in the supplementary to help the readers to better understand the methods and interpret the results.*

You are absolutely right. As long as we used volume-to-biomass ratios, the fit is the same for AGBD or WVD except for a scaling factor (which is different for coniferous and broadleaved forests). For this reason, the way we obtained FORMS-V was to convert FORMS-B to WVD with these volume-to-biomass ratios. We did not write it as it was equivalent to fitting a FORMS-H and NFI WVD relationship. We will modify the manuscript to clarify this point.

*6)      Fig. 4b, it seems that the generated canopy height in the third column is not well matched with Google map, any reasons for that?*

We suppose here that you talk about the third row rather than the third column. We are unsure here what you mean by "not well matched". The main delineations of the fields and forests are visible, and the generated canopy height matches the image from Google Maps. However, there is a time mismatch between the Sentinel-2 images (2020) used to produce the canopy height map and the Google Maps images (2019) used here for visualization purposes. Some forest parcels on the middle-left part of the image must have been clear-cut between the two satellite acquisitions. We will add the dates of the Google Maps images (2020, 2018, and 2019) in the caption of the figure to warn readers of time mismatch.

*7)      Fig. 6, why select those four regions for comparison? What's the model performance across the entire ALS dataset? Does the generated dataset still outperform other products?*

These four regions were selected only to interpret the comparison between the products visually. However, the scatterplots and error metrics were calculated for the entire ALS1 and ALS2 datasets, showing that the generated dataset tends to be better than other products. We acknowledge here that this needed to be stated clearly, and we will modify the caption of Fig. 6 to clarify it.

*8)      Fig. 7, similar problem to my comment#6*

Thank you for this comment. As FORMS-B is derived from FORMS-H, we observe the same phenomenon here. Please refer to the answer to comment #6 for further details.

*9)      Fig.8e-f, do the data points represent the AGBD data across all sites in GLORIE and Renecofor or only represent sites falling into selected regions of Fig.8a-d? Please clarify it in the figure caption.*

As for comment #7, we acknowledge that it must clearly be stated that the scatterplot corresponds to the whole dataset. We will modify the caption of Fig.8 to clarify this point.

*10)     Fig.9, what about the R2 metric for the comparison?*

Thank you for this comment. We will add it to the modified version of the manuscript.

*11)    Any potential limitations for the generated dataset so that the readers can further improve it?*
We can raise the following limitations and potential improvements for further studies: The quality of FORMS-H is lower for steep slopes, as shown by the comparison with ALS 2. Further studies, specifically focused on mountainous areas, are needed to improve the accuracy in these regions. Additionally, our deep learning model is trained only on composite S1 and S2 images for 2020. Applying the same model to other years may lead to significant errors, as composite images would look different. It would be interesting to have a year-agnostic model that is able to derive a canopy height map from S1 and S2 images from any year. The creation of FORMS-B relies on two height-biomass power-law allometric equations for broadleaved and coniferous forests. Even though this very simple approach yields acceptable results for biomass, there is still a significant difference against validation data. In the future,  one could consider deriving plot-level allometry by tree species/ecoregions to obtain a more precise biomass estimation. This would involve using tree species maps and enough NFI AGBD samples for each species. Introducing other predictors than height to infer biomass could also be interesting. It is well established that dominant tree height alone is not evidently related to biomass because of tree cover, density, saturation of height at high biomass, and understory trees. For instance, maps of tree cover obtained from GEDI may help the retrieval of forest biomass, especially in forests where tree cover is less than one.
We will add text about limitations and possible ways to overcome them at the end of our revised manuscript.

*12)    The title and main text contain FORMS-H, FORMS-V and FORMS-B, but the abstract only showed the results of FORMS-H and FORMS-B. Briefly introducing the performance of FORMS-V is therefore needed to show the quality of the generated dataset.*
Thank you. We will introduce the performances of FORMS-V shown in Fig. 9 in the abstract, as suggested.

---

## Author Comment (AC2)

**Answers to RC2**

*RC2 original text is italicized and grey*. Our answers are in plain text and black

*I am happy to read this manuscrip from Schwartz et al. This manscript developed canopy height, wood volue density, and aboveground biomass density data products in France using GEDI, Sentinel-1 and Sentinel-2 datasets with a deep learning approach. The developed data products were assessed with multiple independent datasets and showed improvements over previous developed data products. Overall, this study is well organized and the data products are needed in time to support forest structure and carbon assessment in facing climate change. I only have minor comments.*

Thank you for this positive description of our work. We will do our best to answer your comments in the revised manuscript, as explained in the following.

*1. Abstract may also include the FORMS-V, which is one of the three data products develioed in this study.*

Thank you for this suggestion. FORMS-V will be added to the revised version of the manuscript.

*2. Table 1. "In this study" should be "This study".*

You are right. We will modify the revised version of the manuscript accordingly.

*3. Figure 1. The rasterization of 25-m GEDI footprints to 10m grid may introduce some uncertainties to the model and data products. Is there a way to reduce these uncertainties, for example, using more data quality control or GEDI footprints in pure landscape types?*

Thank you for this very relevant comment. In answer to reviewer #1, we acknowledged that this method could introduce some uncertainties. Still, we showed that rasterizing GEDI on a 10 m grid provided better results than rasterizing at a lower resolution, like 20 m. You can refer to it for additional details. As you suggested, several techniques could be used to control the quality of GEDI data and help reduce these uncertainties. We already applied basic filters to the GEDI footprints (e.g., quality flag =1), but other filters could have been used to refine the quality of the footprints, such as taking the night GEDI shots and the full power beams only. However, we assumed here that our deep learning model would be able to cope with these uncertainties, and we chose to keep as many footprints as possible.

We also thought of filtering GEDI data based on landscape types, as suggested. It would avoid the case where a GEDI footprint falls at the border between two landscape types (forest border for instance), thus possibly creating a high uncertainty in the rasterized value. However, our attempts did not yield significantly better results. Moreover, this type of landscape filtration would also remove all GEDI data on isolated trees and hedges, making it more difficult for the model to predict the height of these trees accurately. For these reasons, we finally decided not to use this type of filtration, but it is a relevant matter that should be addressed in further studies. This will be explained in the revised manuscript

*4. Figure 5. What is the reason that the R2 values are so different in Figure 5a and Figure 5b?*

We assume you are referring to the R² values of the figures 5.1.a and 5.2.a. They represent the validation scatterplots between FORMS-H and two validation datasets: the GEDI Test dataset for Fig. 5.1.a (R² = 0.33, MAE = 4.48 m) and the French NFI plots for Fig. 5.2.a (R² = 0.69, MAE = 2.94 m). We can indeed observe here a significant difference between the R² values and also between the MAE values, not totally expected because NFI plots are more independent and should be more difficult to predict than GED Test data. A visual interpretation of the scatterplots, associated with the histograms shown in Fig. 5.1.b and Fig. 5.2.b, reveals that low heights are poorly predicted in the GEDI Test dataset and thus greatly impact the R² value. This is rather due to GEDI label errors than an error from the model as long as it cannot be observed in other validation datasets. This issue was already addressed in the original manuscript in lines 228-233. The text in **bold** in the following has been added or modified to address your comment and bring additional information: "*Conversely, FORMS-H indicates higher heights than the labeled GEDI footprints for many areas categorized as low heights (Figure 5.1.a). This discrepancy can likely be explained by the quality of GEDI data, where the labels could be wrong due to atmospheric conditions or geolocation errors.* **These geolocation errors should normally have a symmetric pattern, with as many points overestimated for lower heights as points underestimated for higher heights. However, as detailed in the figure caption, we plotted only the footprints geolocated in forest pixels of the Copernicus DLT map. Therefore the geolocation errors related to GEDI footprints located outside forests were excluded from this graph.** *Our comparison with the completely independent French NFI data (Fig. 5.2)*  **does not reveal the same outlier pattern because these forest inventory measurements are more reliable and accurately geolocated.** *It yields a smaller MAE of 2.94 m* **and a higher R² of 0.69** *(Fig. 5.2.a) with a distribution of predicted data very close to the NFI distribution of heights (Fig. 5.2.b).*"